# Generalized Design, Modeling and Control Methodology for a Snake-like Aerial Robot

**DOI:** 10.3390/s23041882

**Published:** 2023-02-07

**Authors:** Moju Zhao, Takuzumi Nishio

**Affiliations:** 1Department of Mechanical Engineering, The University of Tokyo, Tokyo 113-0021, Japan; 2Department of Mechano-Infomatics, The University of Tokyo, Tokyo 113-0021, Japan

**Keywords:** aerial robot, snake-like, modeling and control

## Abstract

Snake-like robots have been developing in recent decades, and various bio-inspired ideas are deployed in both the mechanical and locomotion aspects. In recent years, several studies have proposed state-of-the-art snake-like aerial robots, which are beyond bio-inspiration. The achievement of snake-like aerial robots benefits both aerial maneuvering and manipulation, thereby having importance in various fields, such as industry surveillance and disaster rescue. In this work, we introduce our development of the modular aerial robot which can be considered a snake-like robot with high maneuverability in flight. To achieve such flight, we first proposed a unique thrust vectoring apparatus equipped with dual rotors to enable three-dimensional thrust force. Then, a generalized modeling method based on dynamics approximation is proposed to allocate the wrench in the center-of-gravity (CoG) frame to thrust forces and vectoring angles. We further developed a generalized control framework that can handle both under-actuated and fully actuated models. Finally, we show the experimental results with two different platforms to evaluate the flight stability of the proposed snake-like aerial robot. We believe that the proposed generalized methods can provide a solid foundation for the snake-like aerial robot and its applications regarding maneuvering and manipulation in midair.

## 1. Introduction

The development of snake-like robots has a long history that started with the study of real snakes’ motion [1], which provided the mathematical basis. Then, various bio-inspired mechanical designs were introduced to achieve the snake-like configurations [2,3,4,5], and the motion control and planning methods were then proposed for the snake-like locomotion [6,7,8,9,10]. The most unique feature of snake-like robots compared to other legged robots is obstacle-aided locomotion [11], which depends on explicit obstacles for pushing itself and moving forward. During the last decade, underwater snake-like robots have undergone considerable development [12,13,14]. These robots demonstrate a similar mechanism to obstacle-aided locomotion by utilizing the fin structure to generate sufficient reactive forces from the surrounding water.

The origins of terrestrial and underwater snake-like robots can be found in nature. However, it is significantly difficult to find phenomena related to snakes in the aerial domain because snakes have no wings, which are almost mandatory for flight. A unique case of snake motion in the air is via lateral undulation [15], where it uses the body as a wing to act like a glider. However, it is impossible to find a bio-inspired mechanism or motion to enable a snake-like robot to move freely in the air. Thus, we focused on the active propulsion mechanism (that is the propeller) to generate sufficient thrust force to handle the gravity. With this floating ability, the aerial robot can perform snake-like motions, such as squeezing through narrow spaces in midair, which can significantly extend the exploration range in various inspection and rescue applications. Furthermore, the ability of whole body grasping that mimics snake wrapping can be another advantage of a snake-like structure to handle and transport a large object in midair, indicating their superiority over ordinary aerial robots.

In our previous work [16], a unique propulsion system was presented to enable a snake-like aerial robot composed from four links. However, this modular design was not validated with different link numbers, especially with fewer links (e.g., two links) which showed an under-actuated property. Hence, in this work, we first developed a modularized link structure with the proper propulsion system for arbitrary link numbers, and further introduced generalized modeling and control methods to achieve stable flight for both under-actuated and fully actuated models, as shown in Figure 1.

### 1.1. Related Works

#### 1.1.1. Existing Snake-like Robots

Most snake-like robots have a highly modularized configuration, which contains at least one degree-of-freedom (DoF) at a joint to enable the articulated motion. The designs for snake-like robots can be categorized into six groups based on the type of locomotion: (i) robots with passive wheels [17]; (ii) robots with active wheels [18]; (iii) robots with active treads [4]; (iv) robots based on undulations using vertical waves [5]; (v) robots based on undulation using linear expansion [19]; (vi) robots with prismatic joints [20]. Most of the locomotion in these snake robots is obstacle-aided and thus non-holonomic, which indicates that the stability of the whole motion is easy to be achieved due to the large contact surface with the ground. Therefore, the active balance control, which is important for legged robots (particularly the bipedal robot), can be ignored for most of the snake-like robots.

The under water snake: the propulsion system, weak gravity environment and strong environment damping (water resistance). In the water domain, the locomotion depends on the balance between water resistance, buoyancy, and gravity. Thus, a fin structure is generally deployed to enable three-dimensional undulation locomotion in the water [21]. On the contrary, the screw as an active propulsion device is also used to achieve omni-directional locomotion that makes it easier to perform obstacle avoidance and squeeze motion [22]. Given the sufficient damping effect from the surrounding water, the dynamics of underwater robots is relatively slow, which also indicates the easier controllability of the body balance compared to that of legged robots. However, the atmosphere has almost no resistance and buoyancy in most cases, which indicates that aerial robots are required to generate sufficient propulsion forces and a real-time control framework to keep their balance in the air.

#### 1.1.2. Modular Aerial Robots

To achieve the snake-like motion in the air, a modular configuration is also necessary for the aerial robot. The aerial modular structure was first proposed in Distributed Flight Array [23], which introduces a modular flight unit composed of a single propeller. These flight units self-assemble on the ground before takeoff. Although arbitrary assembly shapes are available, the robot cannot change its shape in the air. Then, ModQuad was developed by [24] to enable self-assembly and self-disassembly in the air to provide the opportunity to change shape in the air. However, such a modular robot still has no ability to perform snake-like motion during a flight due to the lack of the joint structure.

Several modular aerial robots with joint connections were also proposed. For instance, a reconfigurable ModQuad connects modular quadrotors in a manner of a closed loop shape [25]. One of the effective applications of this closed-loop shape is to grasp objects using the inside ring structure. However, snake-like motion requires a chained configuration. Then, quadrotors with a chained configuration were proposed by [26], which shows the ability to squeeze narrow space using the snake-like motion. Similarly, a chained aerial robot that was composed of fully actuated flight units was developed by [27], which showed a high freedom of transformation in the air and also demonstrated aerial manipulation with the fixed-root link [28]. A common feature of these robots is that each flight unit has a complete flight ability, since there are more than four propellers in the unit. On the contrary, an articulated aerial robot proposed in [29] has a single propeller in each unit, which indicates the minimum actuators for snake-like motion in the air. However, the motion is only two-dimensional due to the under-actuated property.

To achieve greater maneuvering with joint motion in the air, a unique thrust vectoring apparatus with a dual-rotor was proposed in [16], which resulted in a snake-like aerial robot called DRAGON. This robot has two DoF in each joint module, which enables the highest freedom in both aerial maneuvering and manipulation [30,31]. However, this proposed mechanical design was only validated in four-link type, and the feasibility with fewer links that is an under-actuated model is unknown. Although a similar configuration was applied in the two-link model presented in [32], this robot only showed the stability with the fixed root link, and the flight stability has not been achieved yet. Therefore, the aim of this work was to obtain a generalized design, and modeling and control methods for arbitrary link numbers that can be either under- or fully actuated.

### 1.2. Contributions

Based on our previous works in [16,30,31,32], we propose the generalized methodologies for both under-actuated and fully actuated models that can be summarized as follows:We introduced a generalized design for snake-like aerial robots, including a thrust vectoring apparatus with dual rotors that can generate different thrust forces.We presented a generalized modeling method for articulated aerial robots based on an approximated model and further proposed two different actuator allocation strategies according to the number of vectoring apparatus.We developed a generalized control framework that utilizes the proposed actuator allocation to enable the stable flight for both under-actuated and fully actuated models.We performed experiments with two different platforms, as shown in Figure 1, to demonstrate the feasibility of the proposed design, and modeling and control methods for the flight with joint motion in midair.

We did these things to validate our proposed modeling and control methods, and achieve stable flight for both two and four-link types.

### 1.3. Notation

All the symbols in this paper are explained upon first appearance. Boldface symbols (most are lowercase, e.g., r) denote vectors, whereas non-boldface symbols (e.g., *m* or *I*) denote either scalars or matrices. A coordinate regarding a vector or a matrix is denoted by a left superscript, e.g., ${}^{\left\{A\right\}}r$ expresses r with reference to (w.r.t.) the frame {A}. We define {W} as a unique frame to represent the inertial reference frame. Then, subscripts are used to express: a target frame for a vector or matrix; e.g., ${}^{\left\{W\right\}}r_{\left\{A\right\}}$ represents the 3D position of the frame {A} w.r.t {W}—and/or a relation or attribute for a scalar; e.g., ${}^{\left\{W\right\}}r_{{\left\{A\right\}}_{x}}$ represents the scalar position of the frame {A} along the *x* axis of the frame {W}.

### 1.4. Organization

The remainder of this paper is organized as follows. The generalized design for snake-like aerial robot is described in Section 2. Then, the generalized modeling method is presented in Section 3, which is followed by the generalized control framework in Section 4. Finally, we show the flight experiments using an under-actuated and a fully actuated robot in Section 5. The conclusions are summarized in Section 6.

## 2. Generalized Design

In this section, we propose a generalized design of an articulated aerial robot with distributed rotors, as shown in Figure 2A. Here, the main link configuration of the proposed robot is cylindrical, which is connected by an actuated joint to enable the snake-like motion.

### 2.1. Dual-Rotor Vectoring Apparatus

For a stable flight of this robot, the design of the thrust apparatus is significantly important. We put effort on the design of the minimal thrust apparatus that enables stable flight for an arbitrary number of links, which is based on the following two aspects: the vectoring DoF and the rotor number.

For the vectoring DoF, we consider a 2-DoF vectoring apparatus to achieve the three dimensional thrust force. As shown in Figure 2B, this concept is achieved by equipping two perpendicular vectoring angles ($\phi_{j},\theta_{j}$). This allows the thrust force to point any direction in arbitrary posture and maximize its performance in the flight control. This pair of vectoring axes are controlled by two independent servos with a general PID position control to track the desired angles. On the other hand, $\theta_{i}=\pm 90^{\circ }$ would cause a singularity, because the thrust force can only point along the direction of link rod, regardless of the change in $\phi_{i}$.

Next, we determine the number of rotors on a vectoring apparatus. To achieve the minimal configuration, the number of rotors should be as low as possible. For a single rotor deployment, the optimal rotor position should be $\left\{F_{j}\right\}$, as shown in Figure 2B, which, however, would collide with the vectoring shaft. Furthermore, only a single rotor in each vectoring apparatus (that is in each link module) would also induce the singular configuration that cannot fly, such as the straight-line configuration. Therefore, we use a dual-rotor structure, as shown in Figure 2B, and allow the rotors to generate different thrusts. Then, the articulated robots with distributed rotors can fly in a straight-line configuration with more than two vectoring apparatuses. It is notable that the number of the vectoring apparatuses is not necessarily deployed in all links, and the relative position in each link is also arbitrary. Nevertheless, deployment at the center of the link can provide a better weight balance than other position.

### 2.2. Two-DoF Joint Module

To achieve a snake-like motion, the DoF of the link pose w.r.t. the neighboring link should be more than two, which implies two orthogonal joints are necessary between two links. Therefore, we introduce a composite 2-DoF joint module shown in Figure 2C that consists of two identical single joint structures and an inter connection part. The first joint rotates about the pitch axis that corresponds to the angle of $q_{i\_pitch}$, and the second joint rotates about the orthogonal yaw axis that corresponds to the angle of $q_{i\_yaw}$. Each single joint is actuated by a servo independently. Similarly to the servo for the vectoring apparatus, the general PID position control is used for joint motion.

Two typicals model based on the proposed design are depicted in Figure 3. From the next section on, we will discuss the dynamics and control for those models.

## 3. Generalized Modeling Method

In our work, we address the articulated model with more than two links. To achieve the aerial transformation by such a multilinked model, a comprehensive investigation on modeling is important. In this section, we first describe the approximation method to obtain the simplified multilinked model, and then present the actuator allocation for both under-actuated and fully actuated models.

### 3.1. Approximation Model

As shown in Figure 2, the kinematic model of the proposed aerial robot is composed from a chained link structure. We assume the number of links is $N_{L}$; then the vectoring of joint angles $q\in R^{2(N_{L}-1)}$ can be defined as $q:=\left[q_{1\_\mathrm{yaw}},q_{2\_\mathrm{pitch}},q_{2\_\mathrm{yaw}},q_{2\_\mathrm{pitch}},\cdots \right]$. The thrust vectoring apparatus, as shown in Figure 2B, consists of two vectoring angles ($\theta_{j},\phi_{j}$) and dual rotors that generate two thrust forces ($\lambda_{j1},\lambda_{j2}$). Therefore, there are four control inputs, $\theta_{j},\phi_{j},\lambda_{j1}$, and $\lambda_{j2}$, in each vectoring apparatus, and we developed two different usage patterns for the under-actuated and fully actuated models, respectively. It is also notable that the vectoring apparatus is not necessarily deployed in each link module, as shown in Figure 2A. Therefore, the number of rotors, $N_{r}$, can be different from $2N_{L}$.

Then, the dynamic model of such multilinked model w.r.t. the entire CoG frame {CoG} can be written as follows: (1)$$\begin{array}{ccc} {}^{\left\{W\right\}}{\dot{P}}_{\sum }\left(q,\dot{q},\phi ,\dot{\phi },\theta ,\dot{\theta }\right) & = & {}^{\left\{W\right\}}R_{\left\{CoG\right\}}{}^{\left\{CoG\right\}}f-m_{\sum }g, \end{array}$$(2)$$\begin{array}{ccc} {}^{\left\{CoG\right\}}{\dot{L}}_{\sum }\left(q,\dot{q},\phi ,\dot{\phi },\theta ,\dot{\theta }\right) & = & {}^{\left\{CoG\right\}}\tau , \end{array}$$(3)$$\begin{array}{ccc} M_{J}\left(q\right)\ddot{q}+c\left(q,\dot{q}\right) & = & \tau_{q}+\sum_{i=1}^{N_{r}}J_{r_{i}}^{T}f_{i}+\sum_{i=1}^{N_{s}}J_{s_{i}}^{T}m_{s_{i}}g, \end{array}$$
where the first equation denotes the dynamic motion of the entire linear momentum, which is described in the inertial frame {W}, whereas the second equation denotes the dynamic motion of the entire rotational momentum, which is described in the CoG frame of the entire multibody model (i.e., {CoG}). The third equation corresponds to the joint motion. g is a three-dimensional vector expressing gravity.

${}^{\left\{W\right\}}P_{\sum }$ and ${}^{\left\{CoG\right\}}L_{\sum }$ on the left sides of (Equation 1) and (Equation 2) are the total linear and angular momentum, respectively, which are both affected by the joint angles, vectoring angles, and their velocities, whereas ${}^{\left\{CoG\right\}}f$ and ${}^{\left\{CoG\right\}}\tau$ on the right sides are the total wrench obtained from all vectored thrust forces. The allocation from the vectored thrust forces from this wrench is the key to achieving the flight control, which is described in Section 3.2 in detail.

In Equation (Equation 3), $M_{J}\left(q\right)$ denotes the inertial matrix, whereas $c(q,\dot{q})$ is the term related to the centrifugal and Coriolis forces in joint motion. The symbol "s" stands for "segment" in multilinks. $J_{r_{i}}\in R^{3\times N_{J}}$ and $J_{s_{i}}\in R^{3\times N_{J}}$ are the Jacobian matrices for the frames of the *i*-th rotor and the *i*-th segment’s CoG, respectively. $\tau_{q}\in R^{N_{J}}$ is the vector of joint torque, and $f_{i}$ denotes the three dimensional force generated by each vectored rotor.

The entire dynamics model summarized in (Equation 1)∼(Equation 3) shows the high complexity due to the joint motion, and thus the real-time feedback control based on such a nonlinear model is significantly difficult. Therefore, a crucial quasi-static assumption is introduced in our work to simplify the dynamics; i.e., all the joints are actuated well and slowly by servos ($\dot{q}\approx 0;\ddot{q}\approx 0$). Then, the joint velocity and acceleration can be omitted regardless of the joint motion. Under this assumption, the original dynamic model can be approximated as follows: (4)$$\begin{array}{cc} & m_{\Sigma }{}^{\left\{W\right\}}{\ddot{r}}_{\left\{CoG\right\}}\left(q\right)={}^{\left\{W\right\}}R_{\left\{CoG\right\}}{}^{\left\{CoG\right\}}f-m_{\Sigma }g, \end{array}$$(5)$$\begin{array}{cc} & {}^{\left\{CoG\right\}}I_{\Sigma }\left(q\right){}^{\left\{CoG\right\}}\dot{\omega }+{}^{\left\{CoG\right\}}\omega \times {}^{\left\{CoG\right\}}I_{\Sigma }\left(q\right){}^{\left\{CoG\right\}}\omega ={}^{\left\{CoG\right\}}\tau , \end{array}$$(6)$$\begin{array}{cc} & 0=\tau_{q}+\sum_{i=1}^{N_{r}}J_{r_{i}}^{T}f_{i}+\sum_{i=1}^{N_{s}}J_{s_{i}}^{T}m_{s_{i}}g, \end{array}$$
where ${}^{\left\{W\right\}}r_{\left\{CoG\right\}}$, ${}^{\left\{W\right\}}R_{\left\{CoG\right\}}$, and ${}^{\left\{CoG\right\}}\omega$ are the position, attitude, and angular velocity of the CoG frame calculated based on the forward-kinematics from the root link states (i.e., ${}^{\left\{W\right\}}r_{\left\{L_{1}\right\}}$, ${}^{\left\{W\right\}}{\dot{r}}_{\left\{L_{1}\right\}}$, ${}^{\left\{W\right\}}R_{\left\{L_{1}\right\}}$, and ${}^{\left\{L_{1}\right\}}\omega$) with joint angles q.

Equations (Equation 4) and (Equation 5) still show the properties of the time-variant model because q changes over time and affects both the cog position ${}^{\left\{W\right\}}r_{\left\{CoG\right\}}\left(q\right)$ and the overall rotational inertia $I_{\Sigma }\left(q\right)$. (Equation 6) shows the equilibrium between the joint torque, thrust force, and gravity, which can help us to obtain the desired joint torque from the thrust force. Given that we assume that joints are well controlled by the feedback position control of servos, it is indicated that there is no necessity to perform torque control based on (Equation 6). By ignoring (Equation 6), the whole model for control can be finally considered as a single rigid body by our approximation.

### 3.2. Actuator Allocation

Allocation from the three-dimensional thrust forces $f_{i}$ to the CoG wrench ${}^{\left\{CoG\right\}}w\left(:={\left[{}^{\left\{CoG\right\}}f\;{}^{\left\{CoG\right\}}\tau \right]}^{T}\right)$ provides the connection from the whole body feedback control to the actuators that includes the thrust force λ and vectoring angles θ and ϕ.

The force gap ($\lambda_{j1}-\lambda_{j2}$) from the dual rotors, as shown in Figure 2B, would induce a moment load on the servo that controls the vectoring angle ϕ. Since this vectoring servo should be compact and thus weak, it is considered difficult to dynamically control both the vectoring angle $\phi_{j}$ and the thrust forces $\lambda_{j1}$ and $\lambda_{j2}$ at the same time.

Therefore, we developed two different strategies as follows: (1) a dual-rotor mode that allows different forces for $\lambda_{j1}$ and $\lambda_{j2}$, but the vectoring angle $\phi_{j}$ is constant in control framework; (2) a virtual-single-rotor mode that assigns the same force for $\lambda_{j1}$ and $\lambda_{j2}$ to avoid the moment load on vectoring angle $\phi_{i}$, and thus, we can use $\phi_{i}$ as the control input.

#### 3.2.1. Dual-Rotor Mode

The force and torque related to the *j*-th rotor module can be written as: (7){CoG}fj∗=λj∗{CoG}uj,{CoG}τj∗=λj∗({CoG}u{Fj∗}(q,ϕj)×{CoG}uj+κσj∗uj),(8)=λj∗{CoG}vj∗,
where ∗ denotes the index of the rotor, which is either 1 or 2. $u_{j}$ is the thrust unit normal, κ is the ratio of rotor thrust to its drag, and $\sigma_{j*}$ is the rotational direction of each rotor. ${}^{\left\{CoG\right\}}u_{\left\{F_{j*}\right\}}\left(q,\phi_{j}\right)$ is the rotor position that is affected by the joint angles q and the vectoring angle $\phi_{j}$. Then, the relationship between the target wrench and rotor thrust can be expressed by
(9)$$\begin{array}{cc} & {}^{\left\{CoG\right\}}w=\left[\begin{array}{c} \sum_{j=1}^{{\tilde{N}}_{r}}\left({}^{\left\{CoG\right\}}f_{j1}+{}^{\left\{CoG\right\}}f_{j2}\right)\\ \sum_{j=1}^{{\tilde{N}}_{r}}\left({}^{\left\{CoG\right\}}\tau_{j1}+{}^{\left\{CoG\right\}}\tau_{j2}\right) \end{array}\right]=Q\lambda , \end{array}$$
(10)Q={CoG}u11{CoG}u12⋯{CoG}uN˜r2{CoG}v11{CoG}v12⋯{CoG}vN˜r2,λ=λ11λ12⋯λN˜r2T,
where ${\tilde{N}}_{r}=\frac{N_{r}}{2}$ is the number of rotor apparatus, and ${}^{\left\{CoG\right\}}u_{j*},{}^{\left\{CoG\right\}}v_{j*}$ and $\lambda_{j*}$ correspond to (7) and (8).

#### 3.2.2. Virtual-Single-Rotor Mode

Given that the dual rotors generate the same thrust forces, there is no moment that occurs in the vectoring angle ϕ. Then, we can count the pair of rotors as an integrated rotor that generates a combined uni-directional thrust $\lambda_{j}=\lambda_{j1}+\lambda_{j2}$. In addition, the drag moment and gyroscopic moment can be ideally counteracted. Then, the force ${}^{\left\{CoG\right\}}f_{j}$ and torque ${}^{\left\{CoG\right\}}\tau_{j}$ related to the *j*-th rotor module can be written as: {CoG}fj=λj{CoG}R{Lj}(q){Lj}R{Gi_roll}(ϕj){Gj}R{Fj}(θj)b3,(11)=λj{CoG}uj,{CoG}τj=λj{CoG}u{Fj}(q,ϕj)×{CoG}uj,(12)=λj{CoG}vj,
where $b_{3}={\left[0\;0\;1\right]}^{T}$. Definitions of the frames $\left\{L_{j}\right\}$, $\left\{G_{j}\right\}$, and $\left\{F_{j}\right\}$ can be found in Figure 2B. ${}^{\left\{CoG\right\}}u_{\left\{F_{j}\right\}}$ in (Equation 12) is the position of the frame $\left\{F_{j}\right\}$, which depends on the joint angles q and the vectoring roll angle $\phi_{j}$ because there is an offset from $\left\{G_{j}\right\}$ to $\left\{F_{j}\right\}$, as shown in Figure 2C.

Then, the total wrench in the CoG frame can be given by
(13)$$\begin{array}{cc} & {}^{\left\{CoG\right\}}w=\left[\begin{array}{c} \sum_{i=1}^{{\tilde{N}}_{r}}{}^{\left\{CoG\right\}}f_{j}\\ \sum_{i=1}^{{\tilde{N}}_{r}}{}^{\left\{CoG\right\}}\tau_{j} \end{array}\right]=Q\lambda , \end{array}$$
(14)Q={CoG}u1{CoG}u2⋯{CoG}uN˜r{CoG}v1{CoG}v2⋯{CoG}vN˜r,λ=λ1λ2⋯λN˜rT,
where ${}^{\left\{CoG\right\}}u_{j},{}^{\left\{CoG\right\}}v_{j}$ and $\lambda_{j}$ correspond to (Equation 11) and (Equation 12).

If allocation matrix *Q* in (Equation 13) is full-rank, an arbitrary wrench ${}^{\left\{CoG\right\}}w$ can be achieved by the control input of $\lambda_{j}$, $\phi_{j}$, and $\theta_{j}$, which can be considered fully actuated. If a model has more than two rotor vectoring apparatuses, the full pose control can be achieved by using this allocation mode. However, only two apparatuses imply the bijection between six control input (i.e., $\lambda_{1}$, $\lambda_{2}$, $\phi_{1}$, $\phi_{2}$, $\theta_{1}$, and $\theta_{2}$) and full pose motion SE(3), which can easily result in a control input that exceeds the valid range, especially for the thrust force λ. However, for a model with more than three vectoring apparatuses, there is the redundancy in control input. Therefore, we apply the dual-rotor mode for model with two vectoring apparatuses and the virtual-single-rotor mode for other cases.

## 4. Generalized Control Framework

### 4.1. Common Framework

Based on the approximated model and the allocation strategy proposed in Section 3, we present a common framework for both under-actuated and fully actuated models, as shown in Figure 4, which contains the first part for the pose control in the entire CoG motion, which is followed by the control allocation with the allocation strategies proposed in Section 3.2.

### 4.2. Full Pose Control

For the approximated dynamics (Equation 4) and (Equation 5), feedback control based on a common PID control is applied as follows: (15)$$\begin{array}{cc} {}^{\left\{CoG\right\}}f^{des} & =m_{\Sigma }{}^{\left\{W\right\}}R_{\left\{CoG\right\}}^{T}\left(K_{f,p}e_{r}+K_{f,i}\int e_{r}+K_{f,d}{\dot{e}}_{r}\right)+m_{\Sigma }g \end{array}$$(16)$$\begin{array}{cc} e_{r} & ={}^{\left\{W\right\}}r_{\left\{CoG\right\}}^{des}-{}^{\left\{W\right\}}r_{\left\{CoG\right\}}, \end{array}$$
where $K_{f,*}$ are the PID gain diagonal matrices.

The attitude control follows the SO(3) control method proposed by [33]: (17)$$\begin{array}{cc} {}^{\left\{CoG\right\}}\tau^{des} & =I_{\Sigma }\left(K_{\tau ,p}e_{R}+K_{\tau ,i}\int e_{R}+K_{\tau ,d}e_{\omega }\right) \end{array}$$(18)$$\begin{array}{cc} e_{R} & =\frac{1}{2}{\left[R^{T}R^{des}-R^{desT}R\right]}^{\vee }, \end{array}$$(19)$$\begin{array}{cc} e_{\omega } & =R^{T}R^{des}\omega^{des}-\omega , \end{array}$$
where ${\left[🟉\right]}^{\vee }$ is the inverse of a skew map, and $R:={}^{\left\{W\right\}}R_{\left\{CoG\right\}},\omega :={}^{\left\{CoG\right\}}\omega$ for convenience.

The efficiency of this full pose control ((Equation 15) and (Equation 17)) for the fully actuated model has been validated in [31], and thus we extend the usage to the under-actuated model with a proper control allocation in this work.

### 4.3. Control Allocation

#### 4.3.1. Under-Actuated Model

To stabilize the attitude and altitude for the under-actuated model, a truncated desired wrench can be given by
(20)$$\begin{array}{c} {\tilde{w}}^{des}=\left[\begin{array}{c} {}^{\left\{CoG\right\}}f_{z}^{des}\\ {}^{\left\{CoG\right\}}\tau^{des} \end{array}\right]. \end{array}$$

A truncated allocation matrix $\tilde{Q}$ from (21) is also introduced as follows: (21)$$\begin{array}{ccc} \tilde{Q} & = & \left[\begin{array}{cccc} {}^{\left\{CoG\right\}}u_{z_{11}} & {}^{\left\{CoG\right\}}u_{z_{12}} & \cdots & {}^{\left\{CoG\right\}}u_{z_{{\tilde{N}}_{r}2}}\\ {}^{\left\{CoG\right\}}v_{11} & {}^{\left\{CoG\right\}}v_{12} & \cdots & {}^{\left\{CoG\right\}}v_{{\tilde{N}}_{r}2} \end{array}\right], \end{array}$$
where ${}^{\left\{CoG\right\}}u_{z_{j*}}$ is the third element of ${}^{\left\{CoG\right\}}u_{j*}$.

Then, the desired thrust forces $\lambda^{des}$ can be given by
(22)$$\begin{array}{cc} \lambda & ={\tilde{Q}}^{\#}{\tilde{w}}^{des} \end{array}$$
where ${\left(\;\cdot \;\right)}^{\#}$ denotes the weighted MP-pseudo-inverse. For under-actuated position control in the horizontal directions, the attitude in the roll and pitch directions is generally utilized. However, the proposed under-actuated robot can use both attitude and vectoring apparatus angles for *x* and *y* control. Note that we assume that these angles are sufficiently small, and these angles do not affect the attitude and altitude control.

For the position control, the attitude in the roll and pitch angles can be expressed as follows: (23)$$\begin{array}{ccc} \phi_{d}^{des} & = & k_{\phi }\frac{{}^{\left\{CoG\right\}}f^{des,x}\sin\psi_{d}-{}^{\left\{CoG\right\}}f^{des,y}\cos\psi_{d}}{m_{\Sigma }g}, \end{array}$$(24)$$\begin{array}{ccc} \theta_{d}^{des} & = & k_{\theta }\frac{{}^{\left\{CoG\right\}}f^{des,x}\cos\psi_{d}+{}^{\left\{CoG\right\}}f^{des,y}\sin\psi_{d}}{m_{\Sigma }g}, \end{array}$$
where $\left(\phi_{d},\theta_{d},\psi_{d}\right)$ are the XYZ-Euler angles of the CoG frame orientation.

For the proposed rotor-distributed robots, it is difficult to stabilize the flight in some configurations due to the large moment of inertia. To address this problem, we also use the *i*-th thrust vectoring angles $\theta_{j}^{des}$, $\phi_{j}^{des}$ as follows: (25)$$\begin{array}{ccc} \theta_{i}^{des} & = & \tan^{-1}\left(\frac{-n_{i,y}}{n_{i,z}}\right), \end{array}$$(26)$$\begin{array}{ccc} \phi_{i}^{des} & = & \tan^{-1}\left(\frac{n_{i,x}}{-n_{i.y}\sin\alpha_{j}+n_{i,z}\cos\alpha_{j}}\right), \end{array}$$
where $n_{j}$ is expressed by
(27)$$\begin{array}{ccc} n_{j}\;\; & = & \;\;\frac{n_{va,i}}{\left|\right|n_{va,i}\left|\right|}, \end{array}$$
(28)$$\begin{array}{ccc} n_{va,i}\;\; & = & \;{\;}_{L_{j}}^{C}\;R\left[\begin{array}{c} \;k_{\theta_{j}}\left({}^{\left\{CoG\right\}}f^{des,x}\cos\psi_{d}+{}^{\left\{CoG\right\}}f^{des,y}\sin\psi_{d}\right)\\ k_{\phi_{j}}\left({}^{\left\{CoG\right\}}f^{des,x}\sin\psi_{d}-{}^{\left\{CoG\right\}}f^{des,y}\cos\psi_{d}\right)\\ m_{\Sigma }g \end{array}\right]\;. \end{array}$$

#### 4.3.2. Fully Actuated Model

The complete desired wrench can be summarized as follows: (29)$$\begin{array}{c} w^{des}=\left[\begin{array}{c} {}^{\left\{CoG\right\}}f^{des}\\ {}^{\left\{CoG\right\}}\tau^{des} \end{array}\right]. \end{array}$$

The control objective is to calculate the desired thrust $\lambda^{des}$ and the desired vectoring angles $\phi^{des},\theta^{des}$ from the desired CoG wrench $w^{des}$ from (Equation 29). For a model with more than three rotor vectoring apparatuses (i.e., ${\tilde{N}}_{r}>2$), there is infinite solution of (λ,ϕ,θ) according to (Equation 13). Then, an optimal geometry allocation can be given by
(30)$$\begin{array}{cc} \min_{\lambda ,\theta ,\phi } & {\;∥\lambda ∥}^{2}, \end{array}$$
(31)$$\begin{array}{cc} s.t. & \;w^{des}=Q\left(\theta ,\phi \right)\lambda , \end{array}$$
where *Q* is derived from (Equation 14).

Given that (Equation 31) is a nonlinear constraint for this optimization problem, the computational cost is relatively large, and the calculation time would increase as ${\tilde{N}}_{r}$ becomes larger. Therefore, we developed an iterative solution using a gradient to guarantee the convergence to the optimal (or at least suboptimal) solution. We first follow the method proposed by [34], which utilizes the vectored forces $f_{j}\in R^{3}$ from (Equation 11) as an intermediate variable and further defines a combined vector $F={\left[\begin{array}{cccc} {}^{\left\{L_{1}\right\}}f_{1}^{T} & {}^{\left\{L_{2}\right\}}f_{2}^{T} & \cdots & {}^{\left\{L_{N_{r}}\right\}}f_{{\tilde{N}}_{r}}^{T} \end{array}\right]}^{T}$. Then, the above optimization problem can be rewritten as: (32)$$\begin{array}{cc} \; & \min_{F}\;{∥F∥}^{2}, \end{array}$$(33)$$\begin{array}{cc} \; & s.t.\;w^{des}=\tilde{Q}F, \end{array}$$(34)$$\begin{array}{cc} \; & \tilde{Q}=\left[\begin{array}{cccc} {\tilde{Q}}_{col_{1}} & {\tilde{Q}}_{col_{2}} & \cdots & {\tilde{Q}}_{col_{N_{r}}} \end{array}\right]\; \end{array}$$(35)$$\begin{array}{cc} \; & {\tilde{Q}}_{col_{i}}=\left[\begin{array}{c} E_{3\times 3}\\ \left[{}^{\left\{CoG\right\}}u_{\left\{F_{i}\right\}}\times \right] \end{array}\right]{}^{\left\{CoG\right\}}R_{\left\{L_{i}\right\}}, \end{array}$$
where $E_{3\times 3}$ is a 3 × 3 identity matrix and $\left[\;\cdot \;\times \right]$ denotes the skew symmetric matrix of a three-dimensional vector.

Then, the closed-form for (Equation 32) and (Equation 33) and the desired thrust and vectoring angles can be directly given by: (36)$$\begin{array}{cc} \; & F={\tilde{Q}}^{\#}w^{des}, \end{array}$$(37)$$\begin{array}{cc} \; & \lambda_{j}=∥{}^{\left\{L_{j}\right\}}f_{j}∥, \end{array}$$(38)$$\begin{array}{cc} \; & \phi_{j}=tan^{-1}\left(\frac{-{}^{\left\{L_{j}\right\}}f_{j_{y}}}{{}^{\left\{L_{j}\right\}}f_{j_{z}}}\right), \end{array}$$(39)$$\begin{array}{cc} \; & \theta_{j}=tan^{-1}\left(\frac{{}^{\left\{L_{j}\right\}}f_{j_{x}}}{-{}^{\left\{L_{j}\right\}}f_{j_{y}}sin\left(\phi_{j}\right)+{}^{\left\{L_{j}\right\}}f_{j_{z}}cos\left(\phi_{j}\right)}\right). \end{array}$$

Given the offset from the frame $\left\{G_{j}\right\}$ to frame $\left\{F_{j}\right\}$ in the two-DoF vectoring apparatus, as shown in Figure 2B, the result of vectoring angles ϕ and θ from (Equation 38) and (Equation 39) will change the position ${}^{\left\{CoG\right\}}u_{\left\{F_{i}\right\}}$ in (Equation 35) again. Thus, the results of (Equation 37)∼(Equation 39) will no longer satisfy the constraint (Equation 33) because $\tilde{Q}$ has changed again.

Therefore, we introduce the residual term for (Equation 31) using the results of (Equation 37)∼(Equation 39), and also compute the derivative with respect to λ, ϕ, and θ: (40)$$\begin{array}{cc} \epsilon & =w^{des}-Q\left(\theta ,\phi \right)\lambda , \end{array}$$(41)$$\begin{array}{cc} \delta \epsilon & =J_{w}{\left[\begin{array}{ccc} \delta \lambda & \delta \phi & \delta \theta \end{array}\right]}^{T}, \end{array}$$(42)$$\begin{array}{cc} J_{w} & =\left[\begin{array}{ccc} J_{w/\lambda } & J_{w/\phi } & J_{w/\theta } \end{array}\right], \end{array}$$(43)$$\begin{array}{cc} J_{w/\lambda } & =-Q\left(\theta ,\phi \right)\in R^{6\times N_{r}}, \end{array}$$(44)$$\begin{array}{cc} J_{w/\phi } & =\frac{\partial w}{\partial \phi }-\frac{\partial (Q(\theta ,\phi )\lambda )}{\partial \phi }\in R^{6\times N_{r}}, \end{array}$$(45)$$\begin{array}{cc} J_{w/\theta } & =\frac{\partial w}{\partial \theta }-\frac{\partial (Q(\theta ,\phi )\lambda )}{\partial \theta }\in R^{6\times N_{r}}. \end{array}$$

The partial derivative elements in (Equation 44) and (Equation 45) can be calculated from the multilinked kinematics model.

Our goal is to find a solution of λ,ϕ, and θ to guarantee zero ϵ. Then, we start from the initial values $\lambda_{0},\phi_{0}$, and $\theta_{0}$ calculated from (Equation 37)∼(Equation 39), and perform the following linear iteration with the objective of minimizing $\sum_{j=1}^{{\tilde{N}}_{r}}∥\delta \lambda_{j}{∥}^{2}+∥\delta \phi_{j}{∥}^{2}+{∥\delta \theta_{j}∥}^{2}$ at each iteration: (46)$$\begin{array}{cc} {\left[\lambda_{k},\phi_{k},\theta_{k}\right]}^{T} & ={\left[\begin{array}{c} \lambda_{k-1},\phi_{k-1},\theta_{k-1} \end{array}\right]}^{T}+J_{w}^{\#}\epsilon_{k-1}, \end{array}$$(47)$$\begin{array}{cc} \epsilon_{k} & =w^{des}-Q\left(\theta_{k},\phi_{k}\right)\lambda_{k}, \end{array}$$
where $J_{w}^{\#}$ is the psuedo-inverse matrix of $J_{w}$, and $k\in \left[0,1,2,\cdots \right]$ is the iteration number. The most computationally intensive operation in this iteration process is the calculation of the inverse matrix with a size of the 6 × 6 for $J_{w}^{\#}$, which can be solved instantaneously for real-time control. In most of the cases, it only requires 2 or 3 iterations to get a sufficiently small value of ϵ (i.e., $∥\epsilon ∥∼10^{-6}$). This may be attributed to the initial values $\lambda_{0},\phi_{0}$, and $\theta_{0}$ being relatively close to the convergent solution. We finally define the convergent solution as $\lambda^{des}$, $\phi^{des}$, and $\theta^{des}$.

## 5. Experiments

In this section, we present the development of platforms based on the generalized design, modeling, and control methods, and further demonstrate the experimental results regarding the aerial transformation motion.

### 5.1. Platforms

We developed two different types of platforms: (1) an under-actuated robot with three links and two vectoring apparatuses and (2) a fully actuated robot with four links and four vectoring apparatuses. The main pipes of these robots were made of carbon-fiber-reinforced plastic (CFRP) whose diameter was 25 mm and thickness was 1mm, and the power cables were in these pipes. Thus, these links were connected by a joint unit made of aluminum plates. The onboard system diagram is summarized in Figure 5. The robot had an onboard computer and a main control board to perform the state estimation, flight controller, and motion planning. It is notable that the process of flight control can be divided into two parts: (1) process with heavy computation (e.g., the matrix inversion (Equation 22) and the iteration process (Equation 46)∼(Equation 47)) that performed in the onboard computer; (2) real-time process (e.g., attitude control (Equation 17)) that performed in the main control board. Here, the attitude was estimated using IMU on this board at the rate of 500 Hz, and the estimated error was less than 1${}^{\circ }$. In addition, an external motion capture system, of which the position estimated error was less than 1 mm, was applied in our experiment to obtain the position of the root link. Furthermore, each link was equipped with a small control board called neuron that connected with the main control board via control area network (CAN). The joints connecting these links were actuated by servo motors, of which the measurement error of joint angle was less than 0.1${}^{\circ }$.

#### 5.1.1. Under-Actuated Model

The detailed configuration of the under-actuated platform is depicted in Figure 6. The under-actuated robot was composed of three links (root link, middle link, and end link), and root and middle links were equipped with vectoring apparatuses. The main specifications of this robot can be found in Table 1. The CPU of the onboard PC was an Intel m3-8100Y with quad cores. The joint servo motors were XH430-W350-R (Dynamixel, stall torque: 3.4 Nm), and the vectoring apparatus joints for ϕ and θ axes used XH430-W350-R and XL430-W250-T((Dynamixel, stall torque: 1.5 Nm)), respectively. Here, two vectoring apparatuses were the minimal configuration for the articulated robot flight. Furthermore, the diameter of the three-blade propeller was 5 inch, and the maximum thrust was up to 20 N when the voltage was 25.2 V.

#### 5.1.2. Fully Actuated Model

A fully actuated platform was composed of four link modules with four rotor vectoring apparatus as shown in Figure 7. The main specifications of this robot can be found in Table 2. Furthermore, processes related to modeling, control, and motion planning were all performed inside the on-board compact computer, of which the CPU was an Intel Atom x7-Z8700 with quad cores. The main difference from the prototype developed in our previous work [16] was the propulsion system (i.e., rotor and propeller). We improved the thrust performance by increasing the blades number from 6 to 12. More blades resulted in slower rotation speed. Furthermore, we chose an inner rotor that has a KV rate of 2100 KV, which can significantly suppress the vibration due to the rotor high-speed rotation (i.e., 20,000∼30,000 RPM). Onboard batteries were deployed for the flight. Two LiHv batteries (1300 mAh, 22.8 V) were attached at each link to provide power for a pair of rotors, which enabled the maximum flight time of 3 min.

### 5.2. Flight Experiments

The main goal of the experiments in this work is to verify the feasibility of the proposed generalized modeling and control methods for both under-actuated and fully actuated models that are the crucial foundation to performing complex applications. Therefore, we focus on evaluation of the flight stability and robustness with a given joint motion.

#### 5.2.1. Under-Actuated Robot

Aerial maneuvering with joint motion was evaluated with the under-actuated three-link robot. The configuration was changed during flight, as shown in Figure 8A. The under-actuated flight control framework, as described in Section 4.3.1, was used, and the control parameters used in the experiments are summarized in Table 3. The position control ran at 40 Hz on the onboard processor, whereas the attitude control ran at 200 Hz on the main control board.

Figure 8B–E plot the trajectories of thrust forces, joint angles, position, and attitude errors, respectively. In this experiment, the trajectories of thrust forces did not change significantly during hovering with joint motion, as shown in Figure 8B. Here, the joint trajectories q(t) changed from [0.0,0.0,0.0,0.0] to [0.6,0.0,1.2,0.0], as shown in Figure 8C. During flight, this under-actuated robot achieved a stable configuration, and the position and attitude errors became less than around ±0.2 m and ±0.07 rad. The root mean square errors (RMSE) of the position and attitude are summarized in Table 4. During the configuration change, the flight stability could still be guaranteed, which demonstrated the feasibility of the proposed modeling and control methods for under-actuated model.

To evaluate the flight performance of the proposed snake-like aerial robot in high places, we further conducted an experiment that involved large elevation, as shown in Figure 9A. During this experiment, the robot ascended from the floor to the ceiling at the height of approximately 3.0 m, as plotted in Figure 9B. The position-tracking errors during the whole flight are plotted in Figure 9C; the RMSE were $\left[0.099,0.035,0.058\right]$ m. This result not only verified the stability of the proposed snake-like aerial during the ascending motion, but also demonstrated an efficiency in the elevated terrain that is close to the ceiling.

#### 5.2.2. Fully Actuated Robot

A large-scale aerial maneuvering with joint motion was evaluated with the fully actuated platform. Based on the motion planning method proposed in [30], we designed a joint trajectory that can squeeze a small opening in midair like a snake, as depicted in Figure 10. However, in the actual experiment, we omitted the opening and ceiling with the aim of providing a better visualization for the trajectory tracking, as shown in Figure 11. The fully actuated flight control framework, as described in Section 4.3.2, was used, and the control parameters used in the experiments are summarized in Table 5. Both the motion planning and flight control frameworks were performed on the onboard processor.

Figure 12, Figure 13, Figure 14 and Figure 15 plot the trajectories of the CoG pose, joint angles, rotor vectoring angles, and thrust forces, respectively. The trajectory of the *z*-axis of the CoG motion indicates an ascending motion, whereas other axes show more complex motion. Furthermore, the joints changed in a large range (i.e, $\left[-1.0,1.6\right]$), as shown in Figure 13, indicating the feasibility of complex maneuvering by our proposed robot platform. The maximum joint velocity was 0.31 rad/s. Although this velocity broke the quasi-static assumption that we applied in the modeling method, the flight stability during the joint motion was still guaranteed, which showed the robustness of our flight method for the joint motion in midair. The trajectories of the rotor vectoring angles (ϕ,θ) in Figure 15 and the thrust forces λ in Figure 15 showed the behavior of the outputs from the proposed flight framework, which contains the fluctuations that resulted from the D control term in both (Equation 15) and (Equation 16).

The RMSE of position and rotation control during the whole motion is summarized in Table 6. These results demonstrate the relatively high accuracy of full-pose tracking in the situation that involved joint motion, which indicated the feasibility of the modeling and control methods proposed in Section 3 and Section 4. Regarding the robustness against the external disturbance, we considered the ground effect. During the joint motion, the minimum distance from the floor to the lowest link was less than 0.2 m, which corresponds to the time of 4 in Figure 11. The lowest link had significantly higher interference caused by the downwash compared to other higher links, and this interference can be considered as an external force acting at the lowest link. Nevertheless, Figure 12 still demonstrated a relatively small deviation from the desired trajectory of the CoG pose, which indicated that our proposed control method can guarantee promising robustness against varying external disturbance.

The high trackability for the complex trajectory, as shown in Figure 12 and Figure 13, also indicated the potential to perform snake-like squeezing motion in midair, which can benefit the exploration in tight and elevated terrains, such as inspection between the plant pipelines in high place.

In terms of design, the above experiments demonstrated the feasibility of the proposed link unit equipped with a vectorable dual-rotor for different robot configurations. Although a similar articulated aerial robot proposed in [27] also achieved a snake-like motion in midair, the related link unit was composed of eight rotors, which led a larger size than ours. Therefore, our proposed link unit has a significant advantage in maneuvering in tight terrain that was verified in the experiment of Figure 11.

## 6. Conclusions

In this paper, we first presented a generalized design to achieve the configuration of a snake-like aerial robot where the dual-rotor vectoring apparatus is the key feature. Then, a generalized modeling method for the articulated aerial robot based on an approximated model was presented. According to the number of rotor vectoring apparatuses, under-actuated and fully actuated models were further derived, which resulted in two different allocation strategies. Furthermore, a generalized control framework was developed to support both under-actuated and fully actuated models. Finally, two different platforms were built to perform experiments that involved the joint motion in midair and demonstrated the feasibility of our proposed methods.

An important improvement in the control method is the switch between the dual-rotor and virtual-single-rotor modes, as presented in Section 3. Given the switch can cause a discontinuous change in the rotor’s thrust, the flight may become unstable after the transition. Then, it is necessary to develop an interpolation approach to switch between these two modes smoothly. Another key issue that remains in this work is the feasibility of our design, modeling, and control methods for configuration with more links (e.g., >4). A possible problem with a large link number is the elastic vibration due to the lack of bending and torsional rigidity of the link structure. We will utilize the redundancy of the control input to suppress the elastic vibration according to the additional control method presented in [35]. Furthermore, the self-localization is another crucial challenge for this articulated structure, and sensor fusion using the multimodal sensors distributed in each link can be developed in the future. Last but not least, more evaluations of our articulated aerial platform in the field will be performed in future work to demonstrate the advantages of snake-like structure and motion in the aerial domain. Aside from the snake-like squeezing motion for inspection in the tight and elevated terrains, the whole body grasping that imitates snake coiling can be investigated to achieve large object transportation.

## Figures and Tables

**Figure 1 sensors-23-01882-f001:**
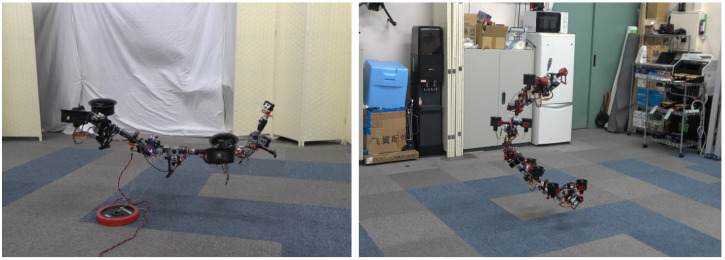
**Snake-like aerial robots with proposed mudular design and generalized modeling and control methods.** (**Left**) Under-actuated model with three links and two thrust vectoring apparatus. (**Right**) Fully actuated model with four links and four thrust vectoring apparatus.

**Figure 2 sensors-23-01882-f002:**
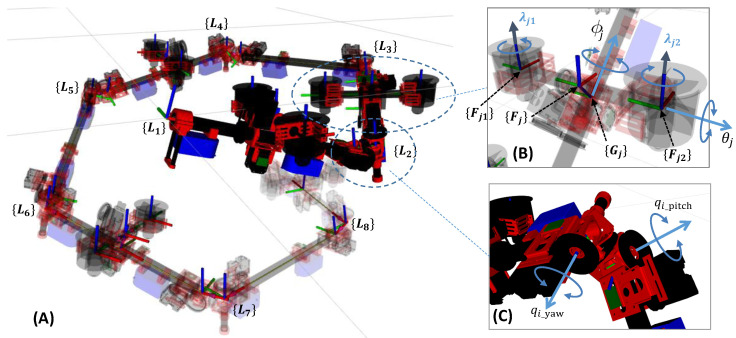
(**A**) Generalized kinematics model of the proposed snake-like aerial robot. $\left\{L_{i}\right\}$ is a frame attached to the start point of the *i*-th link, and the *x* axis is aligned with the direction of link rod. The link does not necessarily contain the thrust vectoring apparatus (i.e., link3, link5, and link 7). (**B**) Two-DoF thrust vectoring apparatuses ($\theta_{j},\phi_{j}$). $\left\{G_{j}\right\}$ is a frame attached to the origin of vectoring apparatus, and the *x* axis is aligned with the *x* axis of $\left\{L_{i}\right\}$ and rotates around it with $\phi_{j}$. $\left\{F_{j1}\right\}$ and $\left\{F_{j2}\right\}$ are the frames attached to the dual rotors, and $\left\{F_{j}\right\}$ is a frame in the middle of the them, where the *z* axis is parallel to the rotor rotation axis and is titled from the *z* axis of $\left\{G_{j}\right\}$ with $\theta_{j}$. $\lambda_{j1}$ and $\lambda_{j2}$ are the thrust forces generated by the dual rotors. (**C**) Two-DoF joint module composed of two orthogonal joint axes ($q_{i\_\mathrm{yaw}},q_{i\_\mathrm{pitch}}$).

**Figure 3 sensors-23-01882-f003:**
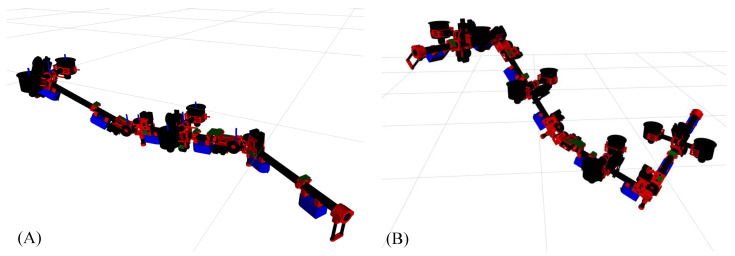
(**A**) Model equipped with three links and two vectoring apparatuses. (**B**) Model equipped with three links and two vectoring apparatuses.

**Figure 4 sensors-23-01882-f004:**
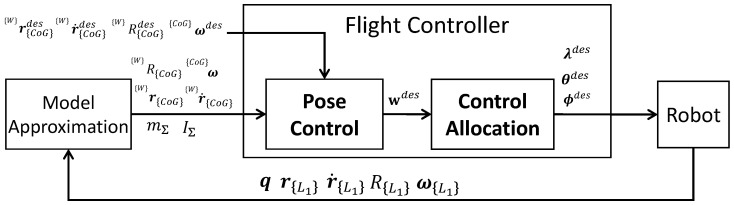
**Overview of the control framework developed in this work, which is a part of the whole system.** “Model approximation” is presented in Section 3.

**Figure 5 sensors-23-01882-f005:**
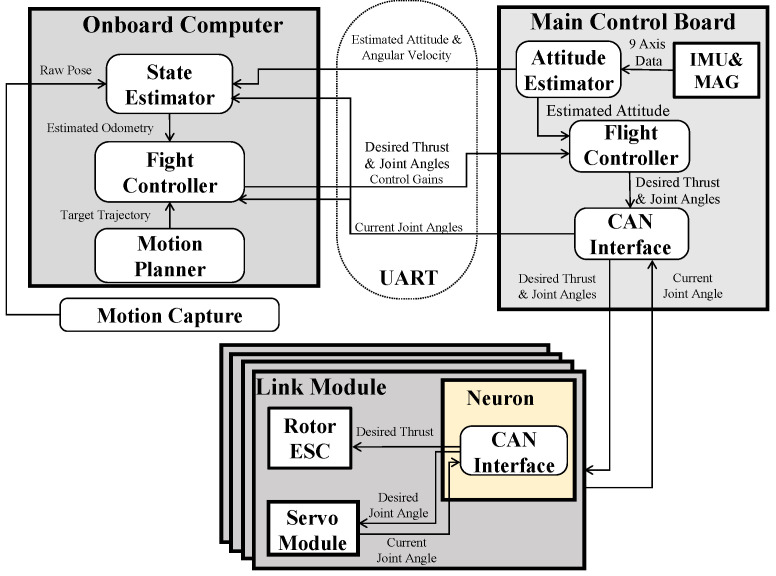
**The onboard system diagram of the proposed snake-like aerial robots**. The system is composed of three parts: (1) an onboard computer to perform the processes that require extensive computational resources; (2) main control board to perform real-time processes, such as the attitude estimate and control; (3) a neuron in each link module to transmit actuator commands from the main control board.

**Figure 6 sensors-23-01882-f006:**
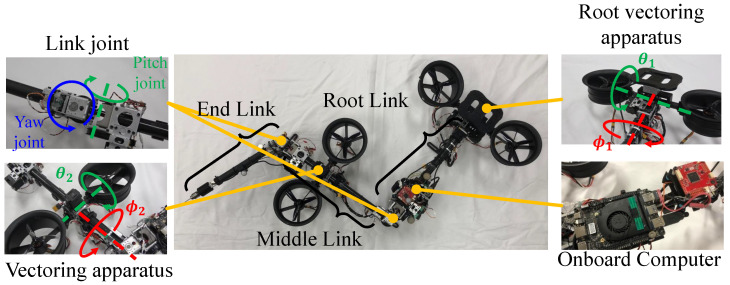
**Under-actuated robot platform with distributed rotors.** This robot was composed of three links and two vectoring apparatuses, and the root and middle links were equipped with the vectoring apparatus. The two rotors on the vectoring apparatus generated different levels of thrust.

**Figure 7 sensors-23-01882-f007:**
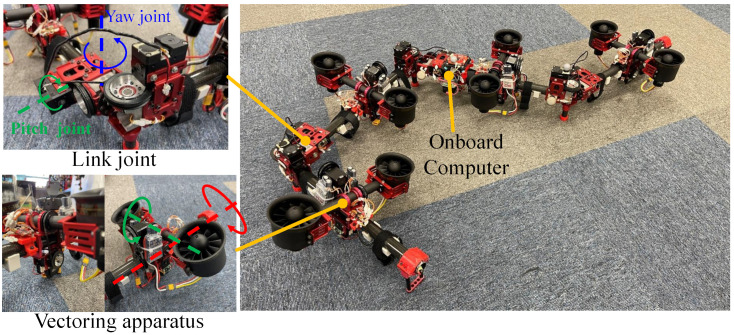
**Snake-like aerial robot composed of four links and four rotor vectoring apparatuses**. The link module contains two orthogonal joint units, and the dual-rotor vectoring apparatus with compact ducted fan rotors is deployed in each link.

**Figure 8 sensors-23-01882-f008:**
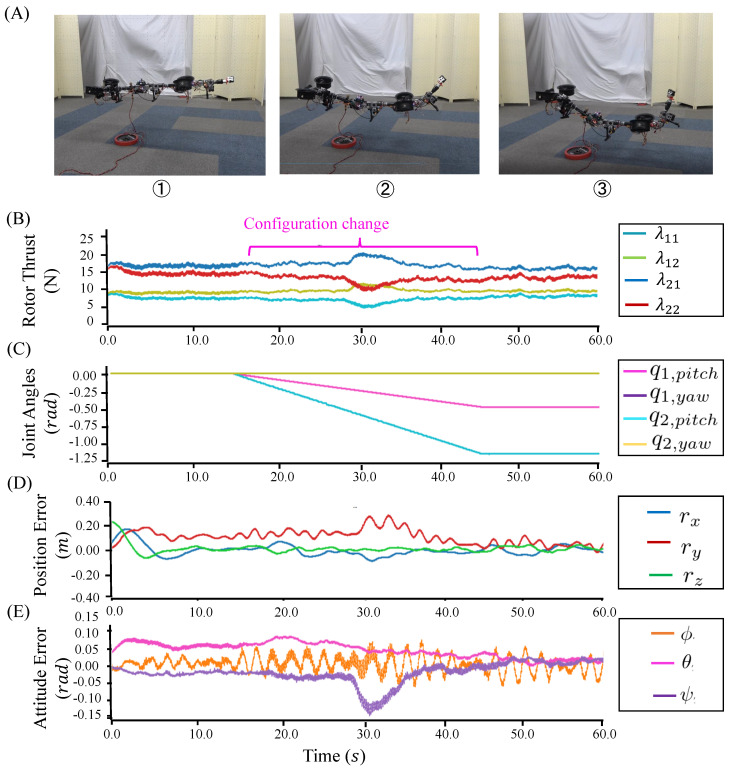
(**A**) Flight of the under-actuated model that involved the configuration change. (**B**) Trajectories for the thrust forces λ. (**C**) Trajectories for the joint motion. (**D**) Position-tracking errors. (**E**) Attitude-tracking errors.

**Figure 9 sensors-23-01882-f009:**
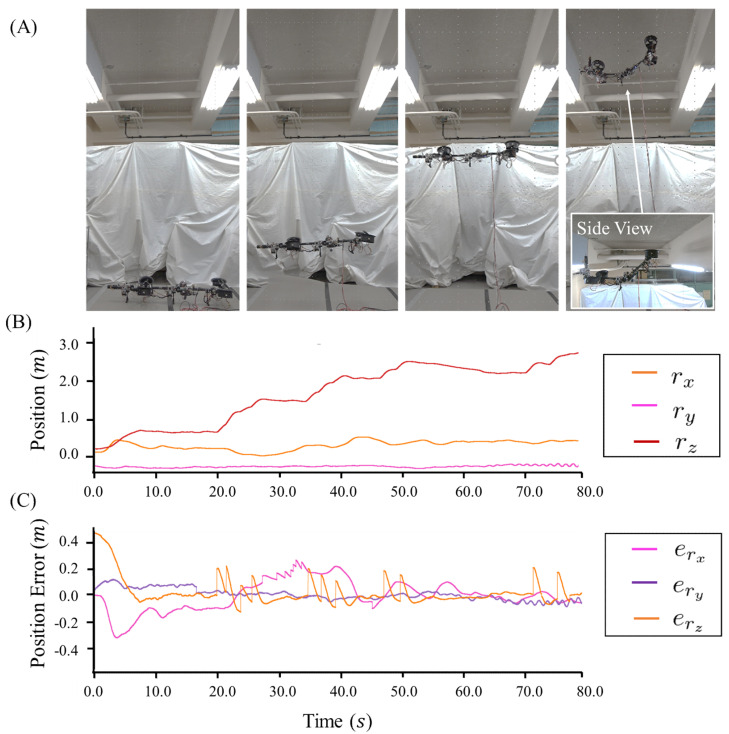
(**A**) Ascending of the under-actuated platform from the floor to the ceiling. (**B**) Trajectories for the CoG position during the flight. (**C**) Position-tracking errors.

**Figure 10 sensors-23-01882-f010:**
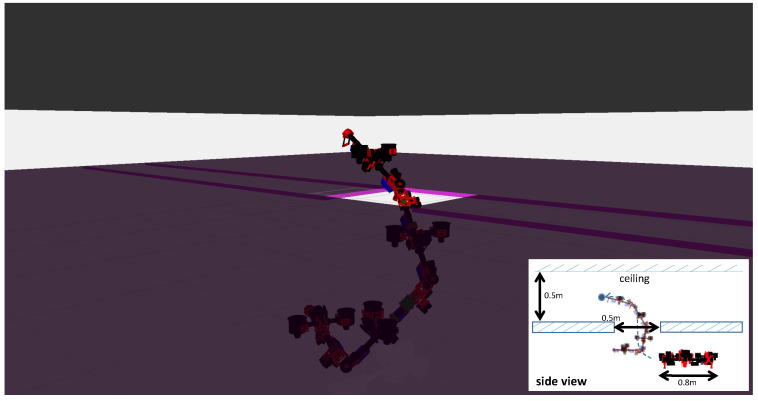
**Joint motion to squeeze a small opening in midair. The width of opening is smaller than that of the robot under the normal form. In addition, a low ceiling also limits the free space. Therefore, a snake-like squeezing motion is required.**

**Figure 11 sensors-23-01882-f011:**
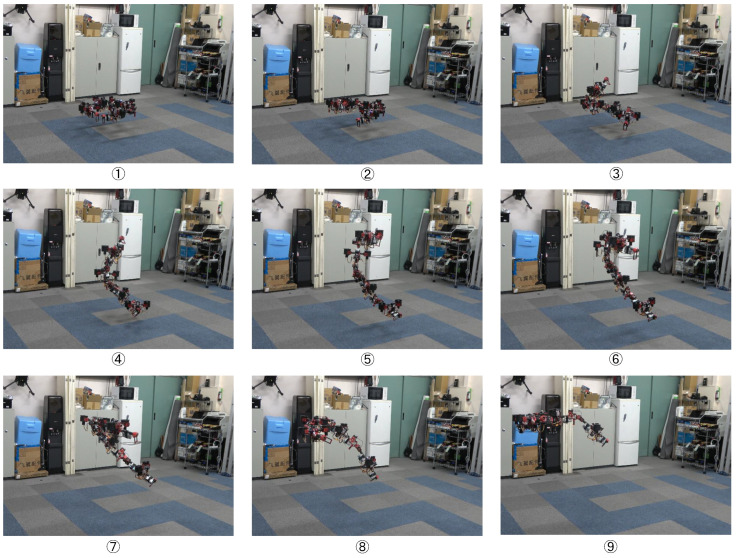
**Complex maneuvering by the proposed snake-like fully actuated aerial robot in midair.**

**Figure 12 sensors-23-01882-f012:**
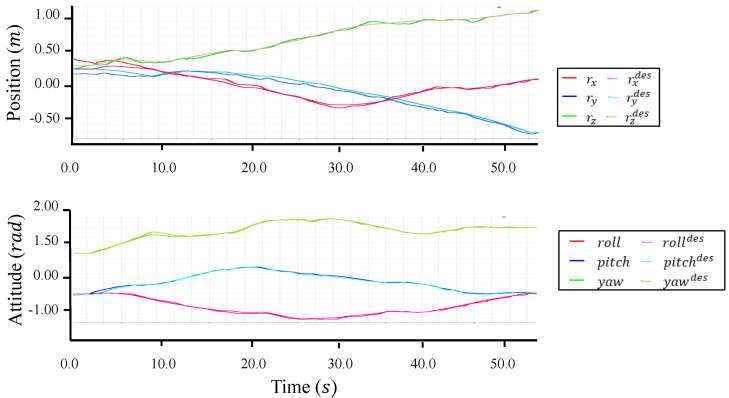
**Trajectories for the CoG pose in the motion of Figure 11**. The comparison between the actual and desired trajectories show the relatively high accuracy of the pose tracking.

**Figure 13 sensors-23-01882-f013:**
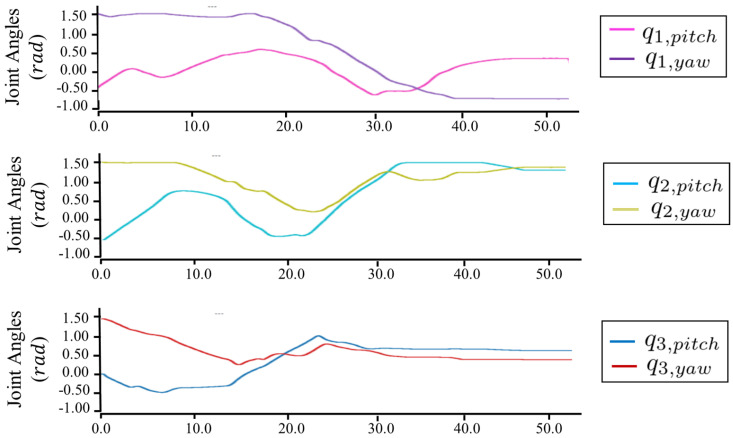
**Trajectories for the joint motion in the motion of Figure 11.**

**Figure 14 sensors-23-01882-f014:**
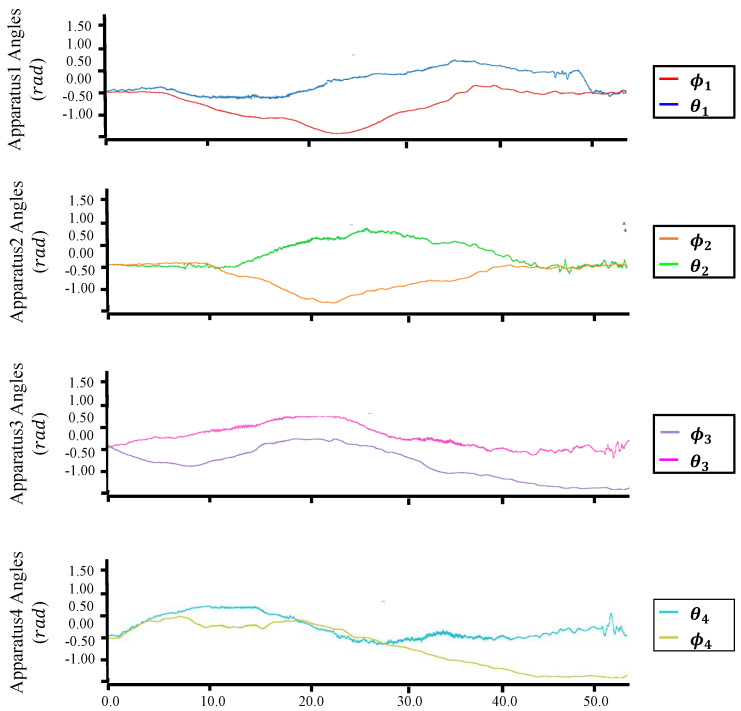
**Trajectories for the rotor vectoring angles in the motion of Figure 11.**

**Figure 15 sensors-23-01882-f015:**
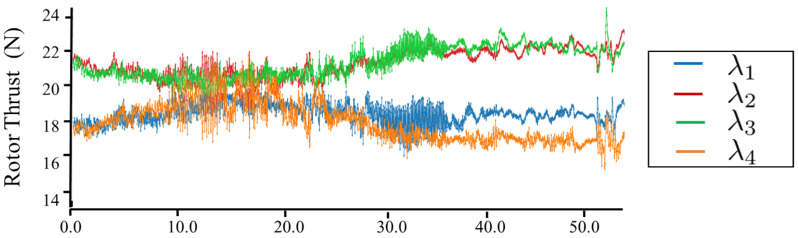
**Trajectories for the thrust force lambda in the motion of Figure 11.**

**Table 1 sensors-23-01882-t001:** Main specifications for the under-actuated model.

Attribute	Description
Root Link	0.40 m
Middle Link	0.40 m
End Link	0.25 m
Total weight	3.35 kg
Propeller Diameter	5 inch
Propeller Blades	3
Max rotor thrust	25 N
Max joint torque	7.0 Nm

**Table 2 sensors-23-01882-t002:** Main specifications for fully actuated model.

Attribute	Description
Link length	0.42 m
Total weight	8.0 kg
Propeller Diameter	70 mm
Propeller Blades	12
Max rotor thrust	32 N
Max joint torque	7.0 Nm

**Table 3 sensors-23-01882-t003:** Control gains for the under-actuated platform.

Parameters	Value
$K_{f,p}$	diag[4.0,3.0,5.6]
$K_{f,i}$	diag[3.0,0.0005,3.0]
$K_{f,d}$	diag[7.0,1.5,3.6]
$K_{\tau ,p}$	diag[1.4,18.0,10.0]
$K_{\tau ,i}$	diag[0.3,0.3,0.3]
$K_{\tau ,d}$	diag[2.0,15.0,6.0]
$k_{\phi },k_{\theta }$	1.0, 0.0
$k_{\phi_{i}}\left(i=1,\;2\right)$	1.0, 0.0
$k_{\theta_{i}}\left(i=1,\;2\right)$	0.0, 0.0

**Table 4 sensors-23-01882-t004:** The RMSE of the proposed under-actuated robot during flight.

	Position (m)	Attitude (Rad)
*x*	0.076	0.033
*y*	0.180	0.063
*z*	0.060	0.062

**Table 5 sensors-23-01882-t005:** Control gains for fully actuated platfrom.

Parameter	Value
$K_{f,p}$	diag(3.6,3.6,2.8)
$K_{f,i}$	diag(0.03,0.03,1.2)
$K_{f,d}$	diag(4,4,2.8)
$K_{\tau ,p}$	diag(15,15,10)
$K_{\tau ,p}$	diag(0.3,0.3,0.1)
$K_{\tau ,d}$	$5E_{3\times 3}$

**Table 6 sensors-23-01882-t006:** The RMSE of the proposed under-actuated robot during flight.

	Position (m)	Attitude (Rad)
*x*	0.036	0.026
*y*	0.038	0.033
*z*	0.017	0.033

## Data Availability

Not applicable.

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
