# Peer review of "Generalized Design, Modeling and Control Methodology for a Snake-like Aerial Robot"

_sensors, 2023, doi:10.3390/s23041882_

Round 1

Reviewer 1 Report

This article proposed the generalized design, modeling, and control methodology for a novel aerial snake-like robot. A thrust vectoring apparatus capable of generating different thrust forces was presented. The authors further introduced the modeling method for two modes. Besides, the full pose control and control allocation were detailed. Then the flight experiments were successfully accomplished. The novelty and contributions are well summarized. I recommend it for publication after addressing the following points:

1) In Sec. 2, the authors claimed their design allows the thrust force to point any direction. Is there any mechanical restriction?

2) In Sec. 3, the dual-rotor and virtual-single-rotor modes are introduced. I wonder if the two modes can be switched freely? Is there any disturbance during the switching process?

3) Some phrases should be uniform throughout the paper. ‘aerial snake-like robot’ or ‘snake-like aerial robot’?

4) The authors are suggested doing a thorough proofread when finalizing the manuscript to correct the typos e.g., in Sec. 1 ‘an active propulsion device is also INTRODUCE to achieve omni-directional locomotion’.

5) The applications and possible challenges of the proposed aerial snake-like robot should be discussed.

Author Response

First of all, we greatly appreciate the valuable feedback. We have taken sincere care of the issues raised by you and thoroughly revised the manuscript. We highlighted the revised description in the manuscript. Please check the response to each comment in the attached note file.

Reviewer 2 Report

1) There are some language problems within the manuscript, please modify the paper carefully. For example, p1, “Ground and Underwater snake-like robots can be found their bio-inspired origins in nature”.

2) For the designed control framework, the paper states that: “For the approximated dynamics (4) and (5), feedback control based on a common PID control is used in [27] as follows”, “The attitude control follows the SO(3) control method proposed by [31]”. Since the controllers in the manuscript are directly from other references, then what are the contributions of this manuscript in this part?

3) The included experimental results are very interesting, and I personally believe the experiment part is the strength of the manuscript. I suggest adding more explanations regarding the experimental setup, for example, how about the sensor part of the systems? What about their accuracy? Also, I think it is more convincing, if some external disturbance is exerted on the system during the test process.

4) More recent reference can be cited to analyze the research state of snake robots, such as “A Reinforcement Learning-Based Strategy of Path Following for Snake Robots with an Onboard Camera, Sensors, 2022, 22, 9867”.

Author Response

(The authors gave the same response as above.)

Reviewer 3 Report

In this paper, a generalized design and control method for a snake-shaped aerial robot was introduced. Experimental results for both under- and fully-operated models are also included in the manuscript. There are several comments on this work.

1.     The authors refer to other research papers on modular (chain) aerial robots. However, 'snake-like' aerial robots are not found in the referenced studies. Just as terrestrial snake-like robots mimic snakes' locomotion and propulsion, a reasonable explanation must be provided that aerial robots must be 'snake-like'.

2.     Similar to the advantages of snake-like underwater robots, snake-like aerial robots may be needed in tight and elevated terrain that is difficult for humans to reach. It seems necessary to reconstruct the experiment to show these advantages.

3.     Given the total number of references, self-citations seem a bit much. In particular, it should clearly show novel problem-solving methods compared to the author's previous work.

4.     The experiment settings and and the explanation of the results should be improved. The verification points of the experiments should be clearly determined. It is also necessary to compare how different the results of the experiment differ from those of other studies.

5.     A comprehensive review of grammar, uppercase and lowercase letters (section titles), SI unit expressions, and various abbreviations is required. Degrees of freedom are usually expressed as DoF or DOF. RMSE stands for root mean square error.

Author Response

(The authors gave the same response as above.)

Round 2

Reviewer 1 Report

The author has carefully revised the manuscript according to my comments. It can be published now.

Author Response

Thank you so much for taking the time to review our paper. We have thoroughly checked the spelling in the paper"